# Effects of Meteorological Factors on Waterbird Functional Diversity and Community Composition in Liaohe Estuary, China

**DOI:** 10.3390/ijerph19095392

**Published:** 2022-04-28

**Authors:** Xiuzhong Li, Qing Zeng, Guangchun Lei, Gongqi Sun

**Affiliations:** 1School of Ecology and Nature Conservation, Beijing Forestry University, Beijing 100083, China; joseph_bjfu@163.com (X.L.); zenchant@gmail.com (Q.Z.); 2Center for East Asian-Australasian Flyway Studies, Beijing Forestry University, Beijing 100083, China; 3Academy of Inventory and Planning National Forestry and Grassland Administration, Beijing 100020, China

**Keywords:** waterbird functional diversity, temporal dynamics, climate changes, fourth corner analysis

## Abstract

Functional trait diversity represents ecological differences among species, and the structure of waterbird communities is an important aspect of biodiversity. To understand the effect of meteorological changes on the waterbird functional diversity and provide suggestions for management and conservation, we selected a study area (726 km^2^) in Liaohe Estuary, located in northeast China. We explored the trends of the waterbird functional diversity changes in response to meteorological factors using fourth corner analysis. Our study demonstrated that temperature was a key factor that impacted waterbird functional diversity in spring, while precipitation had a greater impact in autumn. The population size of goose and duck was positively associated with temperature and negatively with precipitation, while that of the waders (Charadriiformes) showed opposite association trends. Herbivores and species nesting on the bare ground exhibited responses to meteorological factors similar to those of geese and ducks, while benthivores and waterbirds nesting under grass/shrubs exhibited trends similar to those of waterbirds. Waterbirds with smaller bodies, shorter feathers, and lower reproductive rates preferred higher temperatures and less precipitation than other waterbirds. In addition, we observed seasonal variations in waterbird functional diversity. In spring, we should pay attention to waders, herbivores, and waterbirds nesting on the bare ground when the temperature is low. In autumn, waders, benthivores, and omnivores need more attention under extreme precipitation. As the global climate warms in this study area, waterbird functional diversity is expected to decline, and community composition would become simpler, with overlapping niches. Biodiversity management should involve protecting intertidal habitats, supporting benthic macrofaunal communities, preparing bare breeding fields for waterbirds favoring high temperatures to meet their requirements for population increase, and preventing the population decline of geese and ducks, herbivores, and species nesting under grass/shrubs. The findings of our study can aid in developing accurate guidelines for waterbird biodiversity management and conservation.

## 1. Introduction

Waterbird community composition is closely related to environmental influences, including climate and landscape changes, which are predominant factors in animal population dynamics [1]. Climate change has a larger impact on waterbird communities than the landscape, as it can impede their ability to respond to landscape variability [2]. Since the mid-twentieth century, waterbirds have been considered excellent indicators of environmental change because they are sensitive to environmental changes and easy to monitor [3,4]. Various diversity indicators can be used to describe the dynamics of the waterbird community composition, such as species abundance and richness. In recent years, another important indicator that is drawing attention and has become the focus of research is functional structure, also known as functional diversity, which is the value and the range of species and organismal traits that influence ecosystem functioning [5,6].

Functional diversity can reveal patterns that cannot be obtained by observing the taxonomic diversity of the entire ecosystem [7] and can provide an accurate reflection of ecological differences among species based on community traits. Waterbird functional diversity reflects considerable temporal changes annually and seasonally [8], as periodic climate variations alter wetland conditions. Temperature and rainfall can affect food production and distribution, influencing waterbird distribution [9]. For example, increasing temperatures and precipitation are associated with migration outflux, thereby decreasing waterbird species diversity [10]. Thus, higher temperatures result in waders abandoning southern habitats to move north [11]. These migratory habits pose a great challenge to observational studies on waterbirds in a wetland habitat [8]. Although migration has been well-studied from the perspective of species abundance and richness under climate change conditions, the functional diversity of migratory and resident species has received little attention. Therefore, studying temporal changes in waterbird community structure and species responses to climate change contributes to a deeper understanding of community dynamics [12] and the role of diversity on the ecosystem [13].

Long-term monitoring is one of the most appropriate ways to study the impact of climate change on temporal trends of waterbird biodiversity [14]. One-third of waterbird populations worldwide migrate and breed in coastal wetlands (UNEP 2006), which are deeply affected by climate changes such as rising sea levels, aberrant freshwater flow, and varying vegetation distribution [15,16]. Hence, studying the functional diversity of waterbirds is highly representative in coastal wetlands.

The Liaohe Estuary coastal wetland is a northern estuary wetland in China with four distinct seasons and dominant vegetation comprising reeds (*Phragmites australis*) and seepweed (*Suaeda salsa*). It is the largest reed field in the world. The three estuaries, Daliaohe, Liaohe, and Dalinghe, provide abundant water resources, vegetation cover, and food sources, creating an excellent habitat for waterbirds [17,18]. Liaohe Estuary is one of the seven most important wetlands across the Yellow and Bohai seas for waterbird stopover and reproduction on the East Asian–Australasian Migratory Birds Flyway (EAAF), and 290 bird species have been recorded here. Approximately one million birds visit during each migratory season, including nearly 500,000 waterbirds across 142 waterbird species, of which 39 are threatened (Avian monitoring report in Liaohe Estuary, 2019). Waterbirds, therefore, account for 63.8% of the total richness and 2% of the total EAAF abundance (EAAFP, 2020). This wetland performs a key role in and is sufficiently representative for the study of waterbird functional diversity. While most previous studies focus on the effects of landscape patterns on waterbird diversity, studying the impact of climate changes on waterbird functional diversity can supply an accurate scientific basis for biodiversity management and conservation. There are abundant waterbirds and distinguished seasons in Liaohe Estuary. In our previous study, we focused on the response of waterbird diversity and abundance to climatic changes, based on abundance and richness, and found that they positively associated with temperature relative variables and the precipitation relative variables showed opposite influences [19]. In this study, we further study the trends of functional diversity based on waterbird traits and its response to meteorological factors, including quantitative dimensions (families, residence, breeding, diet, and nest sites) and qualitative traits (body, flying ability, foraging ability, and reproduction), which has a vital practical significance in Liaohe Estuary.

To explore the response of waterbirds of different types to meteorological factors and reveal different protective measures among different classifications to combat the extreme weather, we conducted waterbird monitoring to elucidate basic waterbird community dynamics in Liaohe Estuary between 2010 and 2020 and collected data on waterbird functional traits described in the literature. By analyzing the temporal variation of waterbird functional diversity and the relationships between its temporal patterns and meteorological factors in Liaohe Estuary, we aimed to understand the ecological impact on northern estuary wetland, the EAAF, and even the world. The primary objectives of this study were to (1) analyze the annual and seasonal fluctuations in waterbird community composition and functional diversity, (2) identify the meteorological factors associated with the functional diversity of waterbirds, (3) predict the impact of future climate change on waterbird functional diversity, and (4) provide suggestions for conservation and management.

## 2. Study Area

The study area is located in Liaohe Estuary National and Provincial Conservation Reserve (LENPCR) (40°38′56.85′′–41°10′25.87′′ N, 121°35′48.91′′–122°15′48.50′′ E) in northern China, which covers approximately 71,200 ha (Figure 1). Liaohe Estuary is located in the northernmost part of Bohai Bay and comprises an alluvial plain of three estuaries: Daliaohe, Liaohe (also known as Shuangtaihe), and Dalinghe [20]. It experiences a continental, sub-humid monsoon and four seasons that include a hot, rainy summer and cold, dry winter. The annual average temperature in this region is 8.6 °C, annual precipitation is 631 mm, and annual evapotranspiration is 1548 mm [21,22]; this area experiences high and low tides twice per day, representing a typical semi-diurnal tide [23]. Historically, this area was formed by retreating sea levels, leaving flat terrain with abundant water. The region has a single vegetation composition, dominated by two plant species—*P. australis* and *S. salsa* [17,18]. Aquaculture has been rapidly developing since 1990, focusing mainly on crab, shrimp, trepang, and shellfish [20,24]. The abundant water and food resources and the superior reed and mudflat habitats attract many waterbirds to stopover and reproduce every migratory season, making this estuary one of the seven most important habitats in the East Asian–Australasian Migratory Birds Flyway (EAAF). Over the past 11 years, 130 waterbird species, including 39 threatened species (IUCN Red List and List of key protected wild animals in China), have been recorded in the LENPCR, and 22 species have been observed in numbers exceeding 1% of their global population. In particular, the Saunders’s Gull *(**Larus saundersi)* population in the LENPCR is the largest globally, with 11,543 individuals observed in spring 2014, accounting for 69.7% of the global population (IUCN, 2018). Hence, this study area is representative and critical for waterbird conservation.

## 3. Material and Methods

### 3.1. Data Collection

#### 3.1.1. Waterbird Observation

Experienced LENPCR staff and volunteers conducted systematic censuses during the waterbird migration season, i.e., in spring (late February to early June) and autumn (late July to early November) every year from 2010 to 2020. The routes, sites, and participants were relatively constant, and high altitude spots were chosen as the monitoring sites to view the surrounding area and easily obtain precise counts. We conducted approximately 11–12 frequency surveys during each migration season, depending on the weather conditions. Waterbird species were identified using a monocular and binoculars, then confirmed by another observer by using A Field Guide to the Birds of China (John MacKinnon and Karen Phillipps, 2000P). We conducted the surveys among sample sites simultaneously or over a short time to avoid repeated counts among the sample sites. When the flocks were larger than 500, an estimated counting approach of 10, 20, 50, 100, or 500 was applied [25]. The maximum sum of the number of each species at all sites in the region during each survey season represented the count for that season. We observed 130 waterbird species, and their respective abundance in every migrating season were recorded for the entire study period.

#### 3.1.2. Functional Traits

To estimate functional diversity, we assessed ten quantitative and five qualitative variables (Table 1) related to waterbird physiological traits to represent key ecological attributes, including family type, residence type, breeding or non-breeding, diet type, nest type, body mass, body size, wingspan, wing length, tail length, bill length, tarsus length, incubation time, clutch size, and nest size. We obtained these data from the website of Birds of the World, Avifauna of China, A Checklist on the Classification and Distribution of the Birds of China, and Avian Monitoring Report in the Liaohe Estuary Natural Reserve (Table 1). These sources describe the quantity and quality of the resources consumed by the waterbirds [26] and the fitness traits of different species, such as their reproduction strategies [27]. For instance, body mass is related to the waterbird energy requirements, diet is related to ecosystem functions, such as seed dispersal and food web structure, and bill and tarsus length to competitive abilities for food capture, resource acquisition, and allocation strategies [9,28,29].

#### 3.1.3. Meteorological Variables

Weather drives environmental productivity changes, resulting in significant variations in the waterbird functional diversity along the environmental gradient [30]. For example, temperature variations impact waterbird traits differently, resulting in variations in the waterbird community composition [31]. Ten indicators were selected to represent the meteorological variables in this study. We collected these data at the coastal meteorological station (22 km from the study area) and from the China Meteorological Data Service Centre website (http://www.data.cma.cn, accessed on 15 May 2022).

We calculated the following variables using the data collected daily during spring (16 February–15 June) and autumn (16 July–15 November) of each year (2010–2020): annual mean temperature (AMT), mean diurnal range (MDR), seasonal mean temperature (SMT), seasonal temperature range (STR), seasonal cumulative temperature days (SCTD), annual precipitation (AP), seasonal precipitation (SP), maximum seasonal precipitation (SPMax), seasonal mean wind speed (SMWS), and seasonal sunshine duration (SSD).

### 3.2. Statistical Analyses

We conducted multivariate statistical analyses of functional diversity using the PC-ORD 7.0 software. We selected a total of 74 common species for the analyses and excluded the species that occurred fewer than ten times and had a summed abundance of less than 1000 to eliminate the potential effect of rare species.

#### 3.2.1. Functional Alpha and Beta Diversity

We examined the functional diversity and its variation over time using two matrices, M × N and N × L, where M was the number of the sample units, N was the number of species, and L was the number of traits. Qualitative trait variables were converted to quantitative dummy variables to obtain a units-by-trait matrix derived from the product of the units and traits matrix. Furthermore, Shannon’s and Simpson’s alpha diversity indices were analyzed based on the community weighted mean (CWM) calculated by abundance and traits matrixes rather than abundance alone. CWM represents the overall community-level trait values by accounting for the abundance of each species in each season [32]. Rao’s quadratic entropy (Rao’s Q) was calculated based on a cross-product matrix similar to that used in principal coordinates analysis but filled with Gower or Euclidean distances. Functional dispersion (FDis), calculated based on principal coordinates analysis, is a functional diversity index sensitive to species abundance but not to species richness. The Gower distance metric gives more weight to species with low total abundance and ignores joint absences [33]. Shannon’s and Simpson’s indices are the most accepted measures of taxonomic diversity [34], and Rao’s Q and FDis are used to calculate trait distance matrices. Similarities in traits among species represent a low functional diversity, and when no species among the samples share any trait, Rao’s Q value will be equivalent to Simpson’s index [35].

Annual and seasonal beta diversity were analogously calculated for each season per year based on CWM, using the Sorensen dissimilarity to represent the total variation among assemblages. We calculated the annual beta functional using dissimilarity for spring and autumn separately over the years, seasonal beta functional diversity between spring and autumn in the same year, and previous autumn and next spring among successive years. Sorenson distance, which highlights responses that are present in the dataset and retains sensitivity to outliers even in heterogeneous datasets [36], was computed using the following formula:$$\beta_{sor}=\frac{\sum_{j=1}^{p}\left|a_{i,j}-a_{h,j}\right|}{\sum_{j=1}^{p}a_{i,j}+\sum_{j=1}^{p}a_{h,j}}\;,$$
where *h* and *i* are sample units *h* through *i*, *j* is the response of certain traits weighted according to the corresponding species abundance, *p* is the total number of responses, and *a* is the CWM value of certain traits. Here, we generalize the formula to *p* traits to examine differences in the CWM value of trait *j* in sample units *i* through *h* as a proportion of the total sum CWM value of all traits in all migratory seasons.

#### 3.2.2. Changes in Meteorological Factors

We explored the changes in meteorological factors over time using monadic regression analysis and examined the significant and linear fit to predict future climate change. The monadic regression analysis has a better prediction effect than most advanced regression models, especially for small datasets [37].

#### 3.2.3. Relationship between Functional Diversity and Meteorological Variables

The relationships between waterbird functional diversity and meteorological variables were analyzed using the fourth corner analysis [38]. Three matrices, species abundance (M × N), meteorological variables (M × S), and species traits (N × L), were built separately for spring and autumn data to estimate bivariate associations between species traits and meteorological variables, using species abundance as a link. Again, 74 common species were included in the species abundance matrix, and the species abundance and trait matrices were transformed by log(x + 1) to normalize data and eliminate dimensions [38]. We excluded the categorical breeding variable because these traits have only two classifications. The meteorological matrix was maintained as raw data because the fourth corner analysis simultaneously centers and standardizes the environmental variables in PC-ORD. The randomization model of the lottery was adapted to eliminate the associated model among the three matrices, and the *p*-value was adjusted for the false discovery rate according to Benjamini and Hochberg [39].

## 4. Results

### 4.1. Waterbird Functional Diversity Dynamics

A total of 130 waterbird species were recorded in the study area from 2010 to 2020. We analyzed 74 common species and divided them into different categories according to families, residence, reproduction, diet, and nest sites. Figure 2 depicts the overall increasing trend in the entire population observed in spring and autumn. We observed the highest species abundance (328,080) for spring in 2020 and the lowest (51,178) in 2010. The highest species abundance for autumn (476,737) was observed in 2018, and the lowest (36,775) in 2012. From 2010 to 2020, the relative abundance of Charadriiformes sharply increased in both spring and autumn, while Lariformes abundance increased gradually. Ciconiiformes and Gruiformes accounted for a small percentage (average 2.46%) of the total species abundance, whereas Podicipediformes, Anseriformes, and Pelecaniformes accounted for a large percentage (average 29.02%) with a decreasing trend over time. For the different residence types, the most abundant species belonged to the summer visitor and passage categories, whereas the species belonging to the winter visitor and resident categories were less abundant. The relative abundance of the summer visitor and passage category species was significantly less in spring than in autumn. The abundance of breeding waterbirds was lower in spring than in autumn, whereas nonbreeding waterbirds showed the opposite trend. According to different diet types, the most abundant during spring were herbivores and benthivores, while the relative abundances of insectivores, carnivores, piscivores, and omnivores were the lowest. The relative abundances of benthivores and carnivores were the highest in autumn, while those of herbivores, insectivores, piscivores, and omnivores were low; meanwhile, the abundance of herbivores decreased sharply in autumn. The relatively high abundance of bare ground and grasses/shrubs nester species made these the more prevalent nesting sites, while the nesting sites on cliffs, floats, and trees were less prevalent (see Appendix A).

The CWM values of each functional trait decreased annually in spring and autumn (Figure 3), except for bill length, which increased in autumn. The fluctuations in body mass, body size, wingspan, wing length, and tail length were very similar in spring but only fairly similar in autumn. Clutch size was generally much smaller in autumn than in spring. Nest size decreased in both spring and autumn and showed trends dissimilar to those of body size and flying ability traits.

For alpha functional diversity (Figure 4), the trends of Shannon’s, Simpson’s, and Rao’s Q indices strongly fluctuated in spring and autumn over time, and the former two indices showed similar trends, while Rao’s Q and FDis showed different trends from the other indices in both spring and autumn. Shannon’s (2.12–2.23) and Simpson’s index values (0.864–0.885) were far higher than Rao’s Q values (0.048–0.084), while FDis values were between 0.214 and 0.284. The former two indices decreased in spring 2015 and 2020 and autumn 2010, 2015, and 2020, while the latter two indices decreased sharply in spring 2019 and autumn 2011. Moreover, the first three indices were higher in spring than in autumn, while FDis was similar between spring and autumn.

For beta functional diversity, the results of the Sorensen dissimilarity calculated based on CWM were compared in pairs among seasons and years. All values were very similar (0.0–0.3), with the lowest value being 0.02 between autumn 2019 and 2020 and the highest being 0.25 between spring 2010 and 2020. Comparing the data of 2020 to those of other years, spring dissimilarity (range 0.04–0.25, average 0.09) was similar to that in autumn (range 0.02–0.18, average 0.08). Between spring and autumn in the same year (range 0.03–0.19, average 0.10), the lowest value was 0.03 in 2019 and the highest was 0.19 in 2010. The dissimilarity values of spring versus spring (range 0.02–0.21, average 0.07), autumn versus autumn (range 0.03–0.11, average 0.07), and autumn versus spring (range 0.03–0.11, average 0.09) were also observed in successive years. Overall, the beta functional diversity was relatively consistent and exhibited the greatest range between autumn and spring in a single year (see Appendix B).

### 4.2. Meteorological Changes

According to the data recorded at the Yingkou meteorological station from 2010 to 2020, we extracted ten variables representing six aspects of meteorological change (Figure 5). The AMT (Figure 5A) was determined using the maximum value of 10.99 °C recorded in 2019 and the minimum of 9.13 °C recorded in 2010. The annual maximum mean diurnal temperature was 8.69 °C in 2014, and the minimum mean diurnal temperature was 7.55 °C in 2010. The mean spring temperature was 8.14 °C in 2010 and 11.49 °C in 2019, and the mean autumn temperature was 17.66 °C in 2010 and 18.75 °C in 2020. The maximum seasonal temperature range was 45.4 °C in spring 2019 and 40.5 °C in autumn 2019, while the minimum seasonal range was 38.1 °C in spring 2015 and 35.1 °C in autumn 2011 and 2015. Across the years, the spring temperature range was generally higher than the autumn range, except in 2016 (Figure 5B). The cumulative temperature days (Figure 5C) varied from 52 to 78 in spring and 94 to 110 in autumn. Annual precipitation generally declined during the study (Figure 5D), with the lowest value detected in 2014 (408 mm) and the highest (1029 mm) in 2010. Spring precipitation fluctuated, whereas autumn precipitation generally declined over the study period. The maximum daily precipitation in spring was 93 mm in 2012, while the autumn maximum was 162 mm in 2011. The seasonal sunshine duration (Figure 5E) was 791 h in 2010 and 1163 h in 2020 for spring and between 706 h in 2010 and 1172 h in 2020 for autumn. Overall, temperature and sunshine variables showed an increasing trend, precipitation and wind speed relative variables showed a decreasing trend, and cumulative temperature days showed an increasing trend in spring and a decreasing trend in autumn. Some of the variables showed autocorrelations with the years, where the strength of associations varied from very strong (AMT and spring mean temperature) to generally strong (MDR, autumn mean temperature, AP, autumn MWS, spring SD, and autumn SD). However, no other variables showed significant associations (Figure 5).

### 4.3. Relationship between Functional Diversity and Meteorological Factors

The results of the fourth corner analysis demonstrated significant correlations between waterbird traits and meteorological factors, with some seasonal differences (Table 2). In spring, family, diet, and nest site type were strongly associated with AMT, MDR, SMT, SCTD, AP, and SSD (Table 2). No relationship was seen with SCTD or SSD, but a strong relationship existed with SP, and nest sites type did not correlate to any meteorological factor in autumn (Table 2). Thus, most changes in the functional structure of the waterbird community were associated with variables related to temperature and precipitation. Furthermore, as the temperature rose and precipitation decreased over time, the species abundance of different groups (Podicipediformes, Anseriformes, and Pelecaniformes, herbivores, cliff, and grass/shrub) declined in spring, while the abundance of Charadriiformes, benthivores, and bare ground nesters increased. Residence traits were not correlated with any meteorological factor except the resident group. In autumn, the abundance of Pelecaniformes, herbivores, and omnivores declined, whereas the abundance of Charadriiformes, Lariformes, piscivores, and benthivores increased. Moreover, abundance of benthivores were negatively correlated with SPMax, and residence type was not significantly correlated with any meteorological factor in autumn.

All physiological traits except for bill length and foraging ability were negatively correlated with temperature and positively correlated with precipitation in spring, but only mass, size, and clutch size showed similar correlations in autumn. In general, species smaller in mass and size, with shorter wing and tail feathers, shorter incubation durations, smaller clutch sizes, and smaller nest sizes, comprised a greater portion of the waterbird community when the temperature rose and precipitation decreased in spring, while weather showed less influence in autumn.

## 5. Discussion

### 5.1. Trends in Functional Diversity

Spring and autumn waterbird abundance increased during the study period (Figure 2A), especially in wader and gull families (Figure 2B); passage and summer visitors (Figure 2C); insectivore, carnivore, piscivore, and benthivore group members (Figure 2E); and bare ground and grass/shrub nesters (Figure 2F). These trends correspond to the characteristics of the study area, which exhibits abundant reed and seepweed in a tidally flat coastal wetland with frequent seawater and freshwater interactions. Abundant water, food resources, and a protected environment under the reserve management attract an increasing number of waterbirds to roost and breed. However, the relative abundance of both breeding and nonbreeding species in spring fluctuated. The abundance of breeding species in autumn was significantly greater than that of nonbreeding species and notably greater than that in spring of the same year (Figure 2D). Summer visitors and residents bred every spring and migrated at the end of autumn; hence, the relative abundance of breeding waterbirds in autumn was greater than that in spring. This demonstrated a higher incubation survival/reproduction rate for these species in this study area than in other breeding grounds where passage and winter visitor species bred. Alternatively, some waterbirds that did not breed here may have died while migrating to this study area [40].

The functional diversity represented by Shannon’s, Simpson’s, and Rao’s Q indices showed declining trends over the study period, and the index values were higher in spring than in autumn (Figure 4A–C). However, Rao’s Q values (0.048–0.084) decreased to far less than Shannon’s values (2.12–2.23), reflecting a minority in the proportion of total abundance represented by species with unique traits. This was likely caused by a low degree of niche differentiation among individuals within communities, suggesting that the most abundant species were similar, and the competition was more intense in the same ecological space [41]. This was reflected in the classification traits, as the increase in abundance of various aspects of the classification was not equal. Species abundance of Charadriiformes showed a sharp increase, while that of Podicipediformes, Anseriformes, and Pelecaniformes showed a decrease. The increase in abundance of species with a particular trait type was prominent (Figure 2). This trend was also observed in the dissimilarity of beta diversity and decreasing trends in the CWM of physical traits, flying ability, foraging ability, and reproduction.

By contrast, FDis showed a relatively stable trend in both spring and autumn over the study period, except for autumn 2011 (Figure 4D). FDis is highly sensitive to abundance and calculates the mean distance from the multidimensional trait space of individual species to the centroid of all species, accounting for species abundance by shifting the position of the centroid toward the more abundant species and weighting distances of individual species by their relative abundances [33]. An increase in FDis reflects an increase in the mean distance of species to the center of the functional space occupied by the community, suggesting that certain functional species grow significantly faster than others, leading to an increase in the average distance of individual species to the centroid [42]. The results reflected that Lariformes species occupied a relatively large abundance in autumn 2011 than others, including Podicipediformes, Anseriformes, Pelecaniformes, Ciconiiformes, Gruiformes, and Charadriiformes, which were less abundant (Figure 2A). This caused disparities in several aspects (Figure 2B,C,E,F), reducing functional diversity in autumn 2011 (Figure 4D). Accordingly, the CWM value of clutch size decreased in autumn 2011 (Figure 3D).

### 5.2. Responses of Waterbird Functional Diversity to Meteorological Factors

Significant differences in various meteorological variables were found among functional classifications and traits. Species from Lariformes, summer visitor and passage, carnivore and benthivore groups, and bare ground and grass/shrub nesters showed large increases in relative abundance over time (Figure 2, Table 2). Opposite responses to meteorological variables were observed among the Podicipediformes, Anseriformes, and Pelecaniformes populations and the Charadriiformes and Lariformes populations. A temperature increase and precipitation decrease led to a decline in Podicipediformes, Anseriformes, and Pelecaniformes abundance, and an increase in Charadriiformes and Lariformes abundance, while that of Ciconiiformes and Gruiformes remained stable. Similarly, high temperatures and low precipitation caused the herbivore and omnivore populations to decrease and the piscivore and benthivore populations to increase. Bare ground nester abundance increased, while grass/shrub nester abundance decreased. Podicipediformes, Anseriformes, and Pelecaniformes represent swimming birds with open water habitats, while Charadriiformes and Lariformes include wading birds that prefer mudflats [43]. In this study, most species belonging to Podicipediformes, Anseriformes, and Pelecaniformes were classified into herbivore and omnivore groups, whereas Charadriiformes and Lariformes mainly included carnivores, piscivores, and benthivores. Fifteen of the 24 species belonging to Podicipediformes, Anseriformes, and Pelecaniformes preferred nesting on grass/shrub, and 18 of the 39 Charadriiformes and Lariformes species preferred nesting on bare ground. There were 13 breeding species among the 24 Podicipediformes, Anseriformes, and Pelecaniformes species and only 16 among the 49 Charadriiformes and Lariformes species in the study area (Appendix A). The nests on the bare ground occurred as small pits without any grass or branches, while those on grass/shrubs were usually concealed and protected by dead branches and leaves [44]. Higher temperatures can promote evaporation, usually accompanied by reduced precipitation, increasing soil salinity [45] and hindering plant growth in coastal areas [46]. This reduces the occurrence of the necessary habitat conditions for the activity, foraging, and reproduction of Podicipediformes, Anseriformes, and Pelecaniformes species. High temperatures and reduced rainfall result in mudflats and intertidal zone exposure, the preferred habitats for Charadriiformes and Lariformes [47,48]. Decreased precipitation alters the flow patterns of rivers and streams, decreasing benthic macroinvertebrate abundance [11]. Thus, when temperature rises and precipitation with a warmer climate occurs in the future, the populations of Podicipediformes, Anseriformes, and Pelecaniformes are expected to decline, while Charadriiformes and Lariformes abundance is expected to increase. Residence types did not correlate with meteorological factors in spring and autumn, likely due to the small study area.

Furthermore, we found that waterbird functional diversity in physical traits, including body mass, body size, tarsus length, incubation, clutch size, and nest size, were negatively associated with temperature variables and positively associated with precipitation and wind speed (Table 2). Bergmann’s rule also indicates that similar species tend to be smaller in warmer regions [49]. Previous studies have suggested that increasing temperatures associated with climate change may cause a decrease in body size, making temperature a key determinant of animal physiology and ecology. This study did not focus on specific waterbird species, but rather on the functional diversity of the entire waterbird community in this region. As the climate warms, rainfall and wind speed decrease, and the abundance of waterbirds with small physical characteristics and low reproductive capacity will increase, suggesting that Charadriiformes species will better adapt to climate change than Podicipediformes, Anseriformes, and Pelecaniformes species. Additionally, some studies have shown that high temperatures result in the wader community moving north. The Pelecaniformes species carry more viruses than waders and are, thus, more susceptible to infection in warmer temperatures and subsequent death [10,50,51,52]. The Podicipediformes, Anseriformes, and Pelecaniformes species are always larger and more fertile than Charadriiformes species, which are generally smaller than other waterbirds. Therefore, a decrease in Podicipediformes, Anseriformes, and Pelecaniformes populations and an increase in Charadriiformes and Lariformes populations may have led to the reduced body mass and reproduction rates seen throughout the study.

### 5.3. Seasonal Differences in Response to Meteorological Factors

Along with annual changes as the temperature increased and precipitation decreased during the study period, functional diversity and community structure composition of waterbirds also showed large seasonal differences, which were likely caused by seasonal meteorological differences. SCTD and SSD showed correlations with waterbird functional traits in spring, while SP and SPMax were more closely correlated with traits in autumn (Table 2). The main conditions for birds to reproduce and hatch are temperature, sunshine, and food. In the study area, the weather conditions included a large diurnal range and less rainfall in spring and stable temperature, increased rainfall, and even extreme precipitation in autumn (Figure 5D). Relatively high temperatures on one day can affect the probability that clutches are initiated the following days [53]. Prolonged daily sunshine causes a cascade effect on food resources, increasing food availability and reproductive rates and success [54]. This was supported by the correlation between benthivore group abundance and temperature in this study.

The results showed that the nest site types were associated with meteorological factors in spring but not in autumn, likely because waterbird reproduction activity usually occurs during spring and large foraging in preparation for autumn migration occurs in autumn. The abundance of waterbirds in the bare ground nester group was positively associated with temperature and negatively associated with precipitation. The grass/shrub nester group showed opposite trends in spring. This is probably because the ground temperature increased in spring, providing favorable conditions for nesting on bare ground. Waterbirds breeding under grass/shrubs rely on increased rainfall and reduced temperatures to benefit plant growth. These requirements are similar for waterbirds with herbivorous diet type, which favor low temperature and high precipitation both in spring and autumn. Similarly, nest size was negatively associated with temperature and positively associated with precipitation in spring but not in autumn.

## 6. Conclusions

Our study examined temporal changes in waterbird functional diversity and their relationships with meteorological variables from 2010 to 2020. The study findings indicate that, although the species abundance and richness increased over the 11-year period, functional diversity did not show many changes, and considerable niche overlap was noted. This suggests that the addition of new species did not create new functional niches, such as fresh food requirements or nesting sites. Tidal input was finite due to the weak hydrodynamic conditions in Bohai Bay, resulting in drastic reductions in the existing food resources of the wetland, particularly regarding invertebrate abundance [55]. As resource availability decreased and requirements increased, the functional structure of the waterbird communities in the study area remained simple, forming a highly competitive ecosystem with strong resilience but weak resistance. Moreover, different responses to meteorological variations such as temperature and precipitation were key factors among different functional traits, particularly low spring temperature and extreme precipitation in autumn. As the climate warmed in the study area, Podicipediformes, Anseriformes, and Pelecaniformes populations decreased and Charadriiformes population increased. Similarly, herbivores abundance decreased, while benthivore abundance increased, bare ground nester species abundance increased, and grass/shrub nester species abundance decreased. This is thought to have led to an increased inter and intra-species competition. In particular, the number of waders sharply increased in the past 11 years, presenting a challenge for intertidal flat habitat environmental conditions. The management and conservation departments should aim to protect intertidal habitats, encourage benthonic animal propagation, and prepare bare ground breeding fields for waterbirds preferring high temperatures. Conservation efforts should be focused on waterbird species preferring low temperatures, such as geese and ducks, herbivores, and species nesting under grass/shrubs. As the waterbird population increases, our results will supply an accurate guide to make conservation measures for management, relieving the pressure on the environment.

## Figures and Tables

**Figure 1 ijerph-19-05392-f001:**
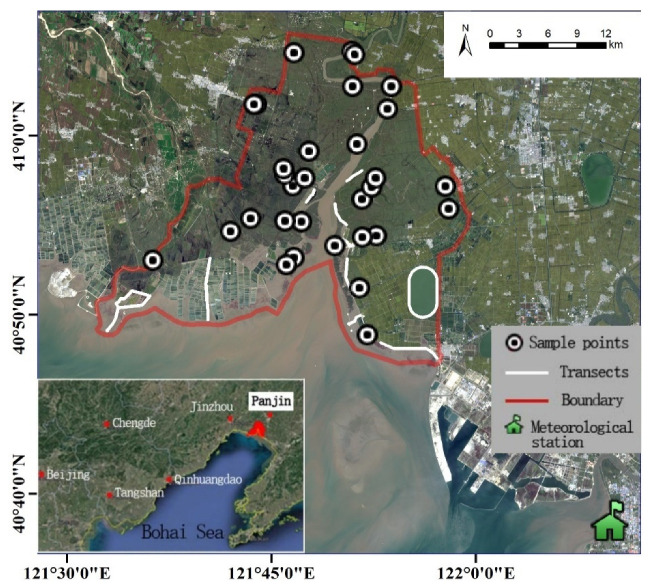
Study area (inside the red border).

**Figure 2 ijerph-19-05392-f002:**
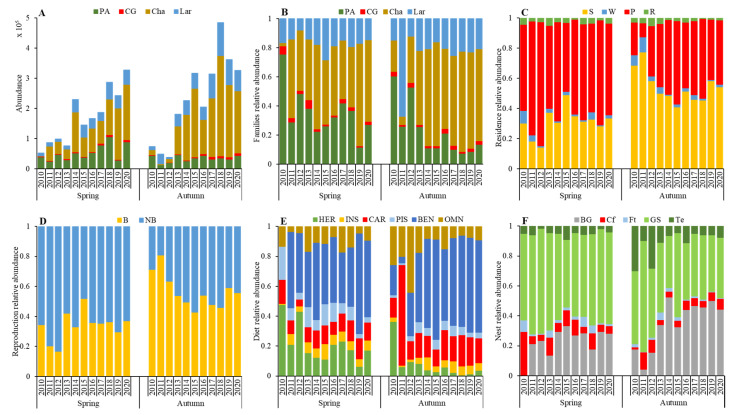
Total abundance of 74 common waterbird species in spring and autumn from 2010 to 2020, and relative species abundance of different categories; (**A**): abundance; (**B**): Podicipediformes, Anseriformes, and Pelecaniformes (PA), Ciconiiformes and Gruiformes (CG), Charadriiformes (Cha), and Lariformes (Lar); (**C**): summer visitor (S), winter visitor (W), passage (P), and residence (R); (**D**): breeding (B) and nonbreeding (NB); (**E**): herbivores (HER), insectivores (INS), carnivores (CAR), piscivores (PIS), benthivores (BEN), and omnivores (OMN); and (**F**): bare ground (BG), cliff (Cf), float (Ft), grass and shrub (GS), and tree (Te).

**Figure 3 ijerph-19-05392-f003:**
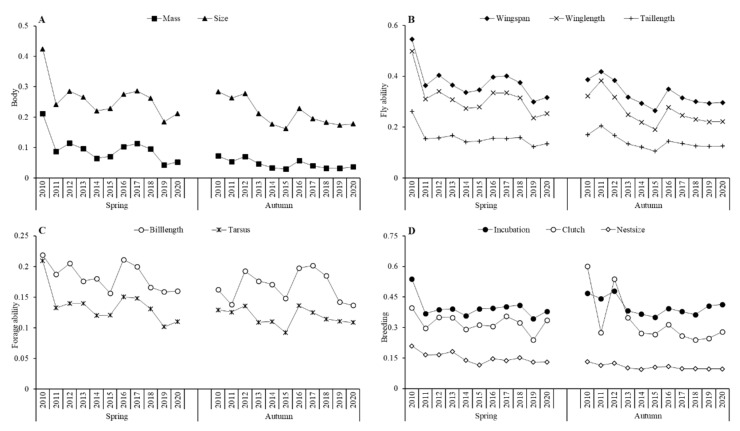
Annual and seasonal trends of community weighted mean (CWM) values: (**A**) Body (mass and size), (**B**) Flying ability (wingspan, wing length, and tail length), (**C**) Foraging ability (bill length and tarsus), and (**D**) Breeding (incubation, clutch, and nest size).

**Figure 4 ijerph-19-05392-f004:**
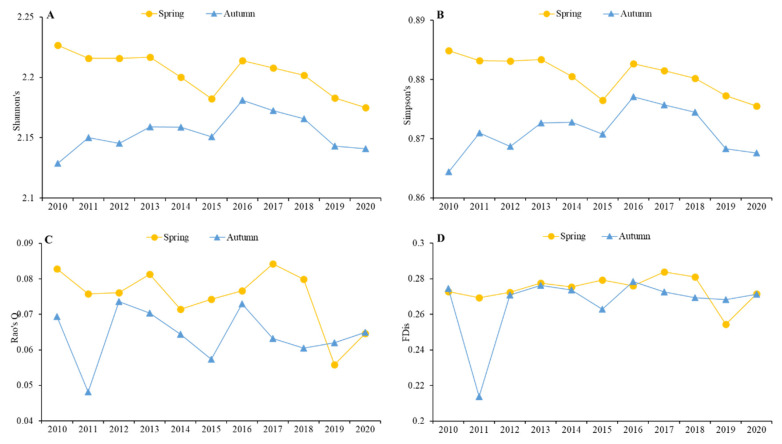
Annual and seasonal trends of alpha functional diversity indexes, (**A**) Shannon’s, (**B**) Simpson’s, (**C**) Rao’s Q, and (**D**) Functional dispersion (FDis).

**Figure 5 ijerph-19-05392-f005:**
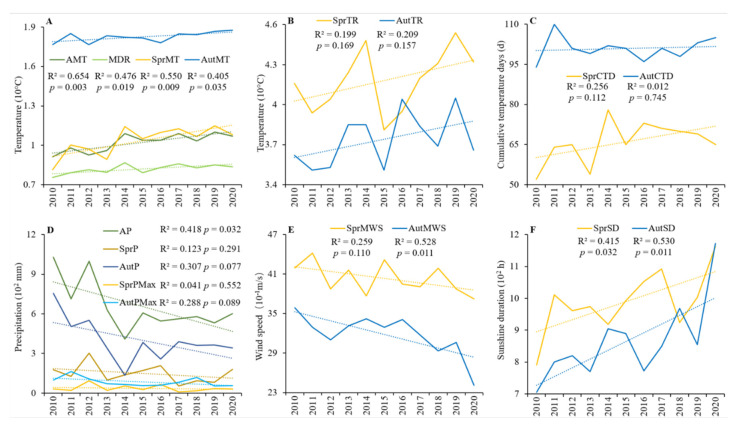
Changes in climatic variables from 2010 to 2020. Variables include (**A**) Annual mean temperature (AT), annual mean diurnal temperature range (MDR); seasonal mean temperature (SprT and AutT); (**B**) Seasonal temperature range (SprTR and AutTR); (**C**) Seasonal cumulative temperature days (SprCTD and AutCTD); (**D**) Annual precipitation (AP), seasonal precipitation (SprP and AutP), maximum seasonal precipitation (SprPMax and AutPMax); (**E**) Seasonal mean wind speed (SprMWS and AutMWS); and (**F**) Seasonal sunshine duration (SprSD and AutSD). “Spr” for spring and “Aut” for autumn.

**Table 1 ijerph-19-05392-t001:** Detailed information on the waterbird variable traits and their descriptions. Categorical variables (C) and quantitative variables (Q).

Categories	Type	Trait	Description	Source
Categories	C	Families	Divided into four groups according to their families and habits	b and c
	C	Residence	Four groups: summer visitor, winter visitor, passage, and resident	c and d
	C	Breeding	Two groups: breeding and non-breeding	b, c, and d
	C	Diet	Six groups: herbivore, insectivore, carnivore, piscivore, benthivore, and omnivore	b
	C	Nest sites	Five groups: bare ground, cliff, float, grass or shrub, and tree	a and b
Body	Q	Mass (g)		a and b
	Q	Size (mm)		a and b
Flying ability	Q	Wingspan (mm)		a and b
	Q	Wing length (mm)		a and b
	Q	Tail length (mm)		b
Foraging ability	Q	Bill length (mm)		b
	Q	Tarsus length (mm)		b
Reproduction	Q	Incubation (day)		a and b
	Q	Clutch (number)		a and b
	Q	Nest size (mm)		a and b

a: Birds of the World, www.birdsoftheworld.org/, accessed on 12 September 2019; b: Avifauna of China, Zhengjie Zhao, 2001; c: A Checklist on the Classification and Distribution of the Birds of China, Guangmei Zheng, 2017; and d: Avian Monitoring Report in Liaohe Estuary Natural Reserve, 2010–2020.

**Table 2 ijerph-19-05392-t002:** Fourth corner analysis of relationships between waterbird traits and meteorological variables in spring (a) and autumn (b). The single symbol represents a significance level of *p* < 0.05, double symbols for *p* < 0.01, triple symbols for *p* < 0.001. “*” is a significant correlation (“+” positive and “−“ negative), “°” is non-significant. PA, Podicipediformes, Anseriformes, and Pelecaniformes; CG, Ciconiiformes and Gruiformes; Cha, Charadriiformes; Lar, Lariformes; S, summer visitor; W, winter visitor; P, passage; and R, resident.

Traits		Meteorological Variables									
		AMT	MDR	SMT	STR	SCTD	AP	SP	SPMax	SMWS	SSD
(a) Spring											
Families		***	**	***	°	**	***	°	°	°	*
	PA	−−−	−−	−−−	°	−−−	+++	°	°	°	−
	CG	°	°	−	°	−	−	°	°	°	°
	Cha	+++	+++	+++	°	+++	−−−	°	°	°	+++
	Lar	°	°	°	°	°	°	°	°	°	°
Residence		°	°	°	°	°	°	°	°	°	°
	S	°	°	°	°	°	°	°	°	°	°
	W	°	°	°	°	°	°	°	°	°	°
	P	°	°	°	°	°	°	°	°	°	°
	R	−	−	°	−	°	+	°	°	°	°
Diet		*	°	**	°	**	**	°	°	°	*
	Herbivores	−−−	−	−−−	°	−−	+++	°	°	°	°
	Insectivores	°	°	°	°	°	°	°	°	°	°
	Carnivores	°	°	°	°	°	°	°	°	°	°
	Piscivores	°	°	°	°	°	°	°	°	°	°
	Benthivores	++	+++	+++	°	+++	−−−	°	°	°	−−−
	Omnivores	°	−	°	°	°	°	°	°	°	°
Nest sites		**	**	***	°	***	**	°	°	°	*
	Bare ground	+++	++	+++	°	+++	---	°	°	°	++
	Cliff	−	−	−−	°	−−	°	°	+	°	−−
	Float	°	°	°	°	°	°	°	°	°	°
	Grass/shrub	−−	−	−−	°	−−	++	°	°	°	°
	Tree	°	°	°	°	°	°	°	°	°	°
Body	Mass	−−−	−−−	−−−	°	−−−	+++	°	°	+	−−
	Size	−−−	−−	−−−	°	−−−	+++	°	°	°	
Flying	Wingspan	−	−−	−−	°	−−	++	°	°	°	−−
	Wing length	−	−−	−−	°	−−	++	°	°	°	−−
	Tail length	−−	−−	−−	°	−−−	++	°	°	°	−−
Foraging	Bill length	°	°	°	°	°	°	°	°	°	°
	Tarsus length	−	−	−	°	−	+	°	°	°	°
Reproduction	Incubation	−−−	−−−	−−−	°	−−−	+++	°	°	°	−
	Clutch	−	°	−−	°	−−	+	°	°	°	°
	Nest size	−−−	−−−	−−−	°	−−	+++	°	°	°	−−
(b) Autumn											
Families		***	**	*	°	°	***	***	°	°	°
	PA	−−−	−−−	−−	−	°	+++	+++	++	°	−
	CG	°	°	°	°	°	°	°	°	°	°
	Cha	++	+	°	°	°	−−	−−	−	°	°
	Lar	+	°	+	°	°	−−	−−	°	°	°
Residence		°	°	°	°	°	°	°	°	°	°
	S	°	°	°	°	°	°	°	°	°	°
	W	°	°	°	°	°	°	°	°	°	°
	P	°	°	°	°	°	°	°	-	°	°
	R	°	°	°	°	°	°	°	+	°	°
Diet		***	***	**	°	°	***	***	**	°	°
	Herbivores	−−−	−−−	−−−	°	−	+++	++	°	+	++
	Insectivores	°	°	°	°	+	°	°	+	°	°
	Carnivores	°	°	°	°	°	°	°	°	°	°
	Piscivores	++	++	+	°	°	−−	−	°	°	°
	Benthivores	++	+	°	°	°	−−	−	−−−	°	+
	Omnivores	−	−	°	−	°	++	++	++	°	°
Nest sites		°	°	°	°	°	°	°	°	°	°
	Bare ground	+	°	°	°	°	−	−	°	°	°
	Cliff	°	°	°	°	+	°	°	+	°	°
	Float	°	°	°	°	°	°	°	°	°	°
	Grass/shrub	°	°	°	°	°	°	°	°	°	°
	Tree	°	°	°	°	°	°	°	°	°	°
Body	Mass	−−−	−	−−	°	°	++	+++	++	°	−
	Size	−−	−	°	°	°	++	++	+	°	°
Flying	Wingspan	°	°	°	°	°	°	°	°	°	°
	Wing length	°	°	°	°	°	+	°	°	°	°
	Tail length	°	°	°	°	°	°	°	°	°	°
Foraging	Bill length	°	°	°	°	°	°	°	°	°	°
	Tarsus length	°	°	°	°	°	+	+	°	°	°
Reproduction	Incubation	−	°	°	°	°	++	++	+	°	°
	Clutch	−−−	−−	−−	−−	°	+++	+++	°	°	°
	Nest size	°	°	°	°	°	°	°	°	°	°

## Data Availability

The data presented in this study are available on request from the corresponding author.

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
