# Peer review of "Effects of Meteorological Factors on Waterbird Functional Diversity and Community Composition in Liaohe Estuary, China"

_ijerph, 2022, doi:10.3390/ijerph19095392_

Round 1
Reviewer 1 Report
General comments
This is an interesting paper about the effect of climate variability on the waterbird functional diversity and community composition in Liaohe Estuary, China. Authors have made a hard fieldwork for collecting such a long-term data and information regarding climate variability, but to be honest, I do not think that just 10 years are enough to evaluate the effect of climate change (although some climate parameters are decreasing or increasing significantly during this 10-years period), especially if one of the main sources of changes in these ecosystems is not taken into consideration, human alterations of water ecosystems by habitat degradation (reducing the surface of water vegetation, eutrophication, pollution, altering water levels…) that can alter habitat and water levels, and then affecting the abundance and diversity of waterbirds.
On the other hand, there is a paper recently published by authors (Li et al. 2021. Waterbird diversity and abundance in response to variations in climate in the Liaohe Estuary, China. Ecological Indicators 132,108286. https://doi.org/10.1016/j.ecolind.2021.108286), not cited in this paper, that use similar methodology, study area and data (from 2010-2019 in Li et al. 2021, and from 2010-2020 in this ms; see figure 2 for abundance and relative abundance graphics in both papers). Although the aims in these two papers are different, this ms is mostly focus on the waterbirds community change during their study period regarding functional traits of waterbirds and two additional functional diversity indexes (Rao’s Q and FDis), these two ms share similar analysis on bird diversity (Shannon’s and Simpson’s functional diversity indexes). Adding one more year of surveys I do not think it is enough to reanalysis your data for a different paper. I think authors have very interesting results regarding the differences of waterbirds assemblages by main diet, nesting behaviour or other functional traits, that are very interesting for a novel paper. So, I recommend authors to rewrite the paper focusing in the assess of differences of climate variability and human habitat alterations on the abundance of waterbirds regarding the functional traits selected by authors in this ms. Moreover, I specifically recommend a revision of the use of the English language.
Reviewer 2 Report
This study aims at assessing the impact of climate on waterbird diversity. However, you seem not to clearly distinguish between weather and climate factors. It is unclear in which case you used weather factors (e.g., if you use average annual temperature to compare to waterbird composition of the given year) and when you used climate factors (multiannual trends). In general, it is difficult to assess the impact of climate in a single study area over a duration of only 10 years, as changes in waterbird composition over time can just be pseudo-correlations not necessarily related to climate. Besides, it seems to me that you only showed a trend for temperature rise in the study area over the 10 years, while there is no clear trend for any directional change in the waterbird composition or functional traits (e.g., fig. 4). In this case, it is very unlikely that any change is driven by climate change. You probably also need to decide if you want to put the focus on seasonal change in waterbird composition or multiannual change (e.g., fig. 3 is a mixture of both, which somehow hinders the reader to see if there is any important result), at the moment, the large amount of little relevant results make it difficult to grasp the most important ones. You need to first decides which these are and then clearly present them. The excessive use of abbreviations makes it difficult to read the text, you should reduce abbreviations to a minimum (if at all). The methods need to be much better described; I could not really comprehend what you did. The language would also benefit from editing, avoid using the passive voice. Finally, I think you should have an objective in your study that you can actually meet using the data you have, I do not think that climate is the right approach. I also wonder if you chose the right journal for your study, how is the study related to public health?
Round 2
Reviewer 2 Report
The manuscript has improved with the revision. However, you need to work more on the language. First, in my opinion there are still by far too many abbreviations, this forces the reader to spend a lot of time to learn them before understanding. Second, the passive voice makes the text awkward and difficult to read, many terms also sound unusual. I strongly recommend to reduce the text length and information provided (better limit it to the essentials) to make the text more readable. Finally, you really need to have the whole manuscript checked by a native speaker.
